# Transverse Thermal Conductivity of Epoxy Carbon Fiber Prepreg Laminates with a Graphite Filled Matrix

**Simon Bard \*** **, Martin Demleitner, Marius Radtke and Volker Altstädt**

Department of Polymer Engineering, University of Bayreuth, Universitätsstr. 30, 95444 Bayreuth, Germany;
Martin.Demleitner@uni-bayreuth.de (M.D.); Marius.Radtke@uni-bayreuth.de (M.R.);
altstaedt@uni-bayreuth.de (V.A.)
\* Correspondence: simon.bard@uni-bayreuth.de

**Abstract:** The thermal conductivity of carbon fiber reinforced polymers is crucial for new technologies and is used in cutting-edge technologies such as sensors, heated rollers and anti-icing of airplane wings. Researchers so far focused on coating conventional prepregs with thermally conductive materials to enhance the transversal conductivity. Another strategy is followed in this study: Thermally conductive matrices filled with graphite platelets were processed by a laboratory prepreg line. Laminates produced from this type of prepregs show an enhancement in thermal conductivity by 3.3 times with a 20 vol% filler content relative to the matrix, and a 55 vol% fiber volume content in the laminate. The research shows that the incorporation of conductive particles in the matrix is more effective for increasing the conductivity than previous methods.

**Keywords:** thermal conductivity; prepreg; carbon fiber; graphite

## 1. Introduction

In the aerospace industry, where lightweight is an important topic to reduce the life-cycle costs of airplanes, metals are increasingly replaced by carbon fiber reinforced polymers (CFRP) [1]. CFRP offer a very high strength to weight ratio, but lack the properties of high thermal and electrical conductivity, which are crucial for some applications. The thermal conductivity is needed to diffuse heat faster, in order to avoid substantial overheating at one side of the part that could lead to microcracking and mechanical defects. Furthermore, the material should conduct the heat from hotspots to a cooling fluid. As explained by Rolfes and Hammerschmidt, the thermal conductivity also influences the mechanical properties, and the calculation of the temperature fields in a part is therefore important for a safe design with CFRP [2].

Only few researchers have so far focused on the thermal conductivity of carbon fiber reinforced laminates. Some publications about the influence of fiber volume content on the thermal conductivity can be found. Rolfes and Hammerschmidt simulated the thermal conductivity with different fiber morphologies, i.e., Polyacrylonitrile (PAN) and pitch based carbon fibers, but the experimental data measured by steady-state guarded hot plate (GHP) only show transverse thermal conductivities for two different fiber volume contents [2]. Pilling et al. prepared laminates with four different fiber volume contents of PAN-based fiber and measured them via transient hot-strip (THS) [3]. The results from Rolfes and Pilling differ significantly, which can be attributed to the different measurement methods used in their work. Very interesting results could also be found by Shim et al., who showed the influence of the fiber shape on the thermal conductivity for pitch-based fiber [4].

One simple strategy to enhance the thermal conductivity is the coating of conventional prepregs with conductive materials. Han and Chung used carbon black, Carbon Nanotubes (CNT) and chopped carbon fiber as a conductive coating. However, the used filler content of up to 0.8 vol% is rather

small. With the guarded hot plate method, they measured an increase in thermal conductivity from 1.091 W/mK to 1.453 W/mK, which is about 33%, with the coating with CNT and a fiber volume content of 65% [5]. Remarkably, the measured conductivity of the unmodified laminate differs from the experimental value of Rolfes, although the fiber volume content is at a comparable level. This can be either attributed to the use of different carbon fiber, voids in the laminates or the different measurement techniques. To compare the results from different researchers, the enhancement in the thermal conductivity in Table 1 is shown in relative values.

The most common approach to enhance the conductivity in epoxy resins without fiber reinforcement is the incorporation of a conductive filler. Carbon-based fillers (graphite [6], carbon nanotubes [7], carbon black [8], and graphene [9]), metallic fillers (Ag, Cu [10], Al [11], and $TiO_2$), ceramic fillers (BN [12], AlN [13], Si [14,15], and $ZrB_2$ [16]) and vegetal fibers [17] can be distinguished.

Promising results could be found with graphene, e.g., a conductivity of 2 W/mK was found in an epoxy matrix with 15 vol% graphene [18]. As known from the literature, the incorporation of nanotubes leads to very high viscosity, which reduces the processability of polymer/nanotube composites [19]. In an elaborative review, Burger et al. concluded that nanofillers may not be suitable to achieve high thermal conductivity, not only because of the difficulty to incorporate higher loadings of filler, but also due to the ineffective heat transport in the matrix [20]. The high number of polymer/nanotube interfaces leads to phonon scattering, reducing the thermal conductivity of the composite material.

Only a single publication dealing with the approach to use conductive epoxy matrices was identified [21]. The authors used boron-nitride as a filler and a mesophase pitch fiber as reinforcement. However, the fiber volume content of the specific samples is not mentioned.

The approach of the underlying research is to improve the thermal conductivity of carbon fiber reinforced composites by the incorporation of filler in the epoxy matrix. First, the effect of graphite as a conductive filler in an epoxy matrix is studied. From the conductive matrix, a carbon fiber reinforced prepreg is produced and processed into laminates.

**Table 1.** Transverse conductivities of samples from the literature and their fiber volume content (FVC) and measurement methods as Laser Flash Analysis (LFA).

| Sample | FVC | Measuring Method | Enhancement Method | Thermal Conductivity Enhancement (%) |
|---|---|---|---|---|
| Epoxy/Carbon Fiber [5] | 65.0 | proprietary | Coating of prepreg: 0.4% SWNT | 33.2% |
| Epoxy/Mesophase Pitch Fiber [21] | n.a. | LFA | Matrix modification | 78.9% |
| Underlying research | 55 | LFA | Conductive matrix: 20 vol% graphite | 330% |

## 2. Production and Characterization Methods

### 2.1. Materials

The resin Tetraglycidylmethylenedianiline (TGMDA, Epikote$^{TM}$ RESIN 496, Hexion Inc., Columbus, OH, USA) is tetra-functional with an epoxy equivalent of 115 g/eq and was cured with Diethyltoluenediamine (XB3473$^{TM}$, DETDA, hydrogen equivalent weight 43 g/eq). Graphite (Imerys Graphite & Carbon, Bodio, Switzerland) with the trade name Graphit Timrex® KS6 and KS44 was used as conductive filler. The filler is characterized by a platelet-shape and an average particle size of 3.6 μm ($D_{50}$) and 18.4 μm ($D_{50}$), according to the manufacturer. The density is 2.255 g/cm$^3$. The particle size distribution was measured and can be found in Figure 1. The data measured in our lab slightly differs from the manufacturer's data. A PAN based fiber 12K A-49 (DowAksa, Atlanta, GA,

USA) with a tensile strength of around 4900 MPa and a Young's modulus of 250 GPa has been used for the prepreg production.

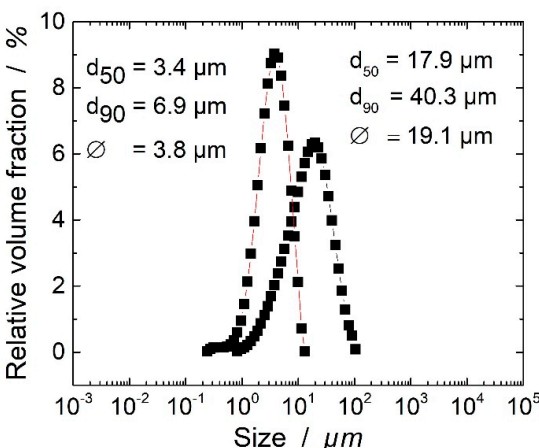

**Figure 1.** Particle size distribution of the used graphite types.

### 2.2. Resin Preparation and Curing

Resin and hardener were stirred at a stoichiometric ratio of 72:28. The mixture was degassed at 10–20 mbar. The mass ratio of graphite is related to the volume of the whole mixture, i.e., to achieve 100 cm$^3$ of the sample with a 15 vol% graphite content, 85 cm$^3$ resin/hardener and 15 cm$^3$ graphite were mixed. Densities of 2.225 g/cm$^3$ for the graphite and 1.2 g/cm$^3$ for the resin were used for the calculations, in order to measure the relevant mass of the resin and filler. A three-roll mill (EXAKT 120EH-450, Norderstedt, Germany) was used to homogeneously disperse the filler in the matrix with opening gaps of 63 and 21 μm. The samples were cured under pressure in a laboratory press at 120, 160 and 200 °C, at each temperature for 1 h with a heating rate of 10 K/min. A postcuring at 220 °C for 2 h followed, before cooling down at a rate of 5 K/min.

### 2.3. Prepreg Production

The unidirectional prepregs were produced via hot-melt processing at the laboratory scale prepreg impregnation machinery of the University of Bayreuth.

First of all, the unidirectional rovings of 12K carbon fibers are pre-spread on the pre-spreading unit, as shown in Figure 2. The resin film was coated at 25 °C on a siliconized carrier paper in the coating unit of the prepreg machinery. Finally, the resin film and pre-spread fibers were impregnated to the final prepreg with a calendar (25 °C). The produced prepregs were then hand-lain up to the final unidirectional prepreg laminates and cured with the same parameter as the neat and filled resin samples. The aimed fiber volume content was set to 55 vol% to ensure the comparability of the prepreg laminates.

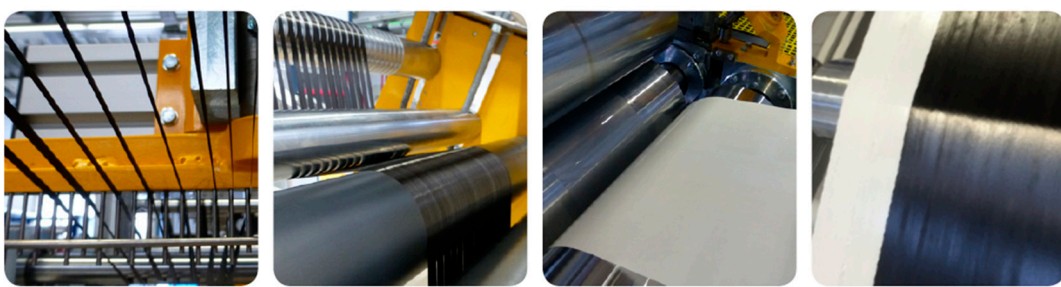

**Figure 2.** Single unidirectional-rovings, pre-fiber spreading unit, resin coating unit and final prepreg, respectively.

### 2.4. Morphological Characterization

All samples were scanned with Skyscan 1072 Micro-CT (Bruker, Artselaar, Belgium), with a linear resolution of 3.50 μm at a magnification of 80 with an accelerating voltage of 80 kV and tube current of 122 μA. The projection images were acquired over 180° at angular increments of 0.23° with an exposure time of 2.57 seconds per frame, averaged over six frames. The three-dimensional images were reconstructed using the reconstruction software provided by the manufacturer (NRecon Version 1.6.4.1, Kontich, Belgium), where the ring artifact reduction was applied as needed.

The samples were sputtered by a Cressington 108 Auto Sputter Coater (Watford, UK) with an Au coating thickness of 13 nm, and they were studied in a Jeol JSM 6510 Scanning Electron Microscope (Tokyo, Japan).

### 2.5. Thermal Conductivity Measurements

The thermal conductivity was measured by the laser flash method (LFA) with LFA447 (Netzsch GmbH, Selb, Germany). Five shots were used with a duration of 30 ms each, and the signal was fitted with the Proteus Analysis Software (Netzsch GmbH, Selb, Germany), using the Cape-Lehman algorithm. The tested samples had a diameter of 12.7 mm. The laser flash analysis is the standard instrument for the determination of thermal transport properties of carbon fiber reinforced polymers because of its convenience, short experiment times and large measurement range [2,22]. The density was measured with AG245 (Mettler-Toledo International Inc., Columbus, OH, USA), using Archimedes' principle. The thermal heat capacity was measured by DSC 1 (Mettler-Toledo International Inc., USA) according to ASTM E1269–11 with a heating rate of 20 K/min.

## 3. Results and Discussions

### 3.1. Neat and Filled Resin Conductivity

For a thorough investigation, first the effects of the filler on the thermal conductivity should be evaluated. This section focuses only on the effects in composites of resin and filler, whereas the next sections focus on the fiber-reinforced composites. The thermal conductivity values of the composites versus the filler content are shown in Figure 3. At lower filler concentrations of up to 10 vol%, the increase in the thermal conductivity is almost linear. At higher filler concentrations, the thermal conductivity increases exponentially. This behavior has already been described by several researchers [20,23,24]. At higher filler concentrations, a remarkable difference in the thermal conductivity can be found between the composites with small and larger particles. In the literature, contradicting influences of the particle size on the thermal conductivity have been reported. Boudenne et al. reported higher thermal conductivities for small copper particles of 12 μm in average in a Polypropylene matrix than for larger particles of 200 μm [25]. Zhu et al. reported contradictory results with boron-nitride of 70 nm and 7 μm in an epoxy matrix. [26] In both studies, it seems that the morphology and shape of the filler was not taken into account. In this study, the shape of the particles is very similar, as can be observed in Figure 4. The aspect ratio was calculated from 60 measurements of the particles observed in Scanning Electron Microscope (SEM) and is at 36 ± 4 for the smaller and 32 ± 5 for the larger particles. The higher thermal conductivity for composites with larger particles can be explained from a thermo-mechanical point of view. Heat is conducted by phonons in polymer composites. It seems that for the thermal conductivity, the interfaces between the conducting parts of the composite are fundamental. As elaborated by Burger et al., the phonons are scattered at the interfaces, which lowers the thermal conductivity [20]. As a result, we can evaluate the resin-filler interfaces and the filler-filler interfaces, in case the particles are in contact with each other. The larger the particle size, the lower the number of interfaces at a given volume fraction of the filler. As already explained by Zhu, the lower number of contacts between particles can explain the better conductivity of larger sized particles [26]. The particle-matrix interaction seems to be more significant than the particle-particle interaction. As explained, the former leads to phonon scattering at the interface.

Smaller particles therefore lead to more interfaces, which result in an increase in phonon scattering and therefore a higher thermal resistance.

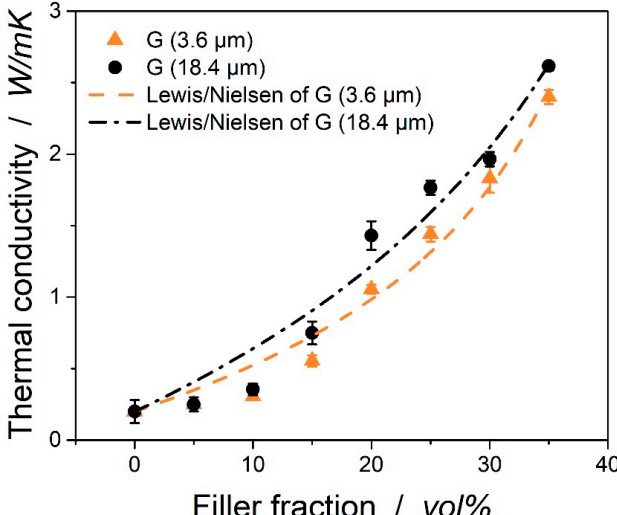

**Figure 3.** Thermal conductivity of epoxy/filler composites depending on the filler content, fitted with equations from Lewis/Nielsen.

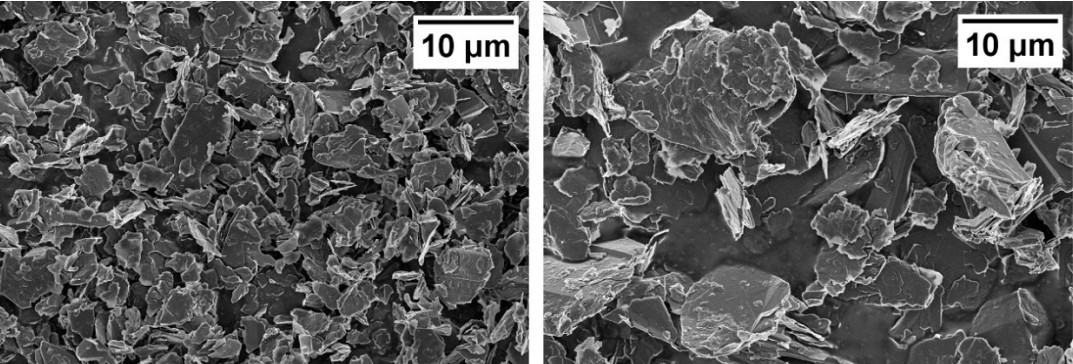

**Figure 4.** Records from scanning electron microscope of small (3.6 μm) and large (18.4 μm) graphitic particles.

Besides the size of the filler shown in Figure 4, the morphology of the samples is crucial for the evaluation of the results from the thermal conductivity. Several methods have been used to study the morphology, from which μCT and SEM seem the most promising. The μCT scans of composites with particles of 3.6 and 18.4 μm are shown in Figure 5. One can clearly see that no voids can be found in the material. The distribution of the filler seems very homogenous. The voxel-size of the μCT is ~2.5 μm, so agglomerations of the micro-sized graphite particles can be excluded. For a deeper investigation of the morphologies, several SEM pictures have been taken. Figure 6 shows two fracture surfaces of composites with 3.6 and 18.4 μm graphitic platelets. Both confirm the homogenous distribution in the matrix.

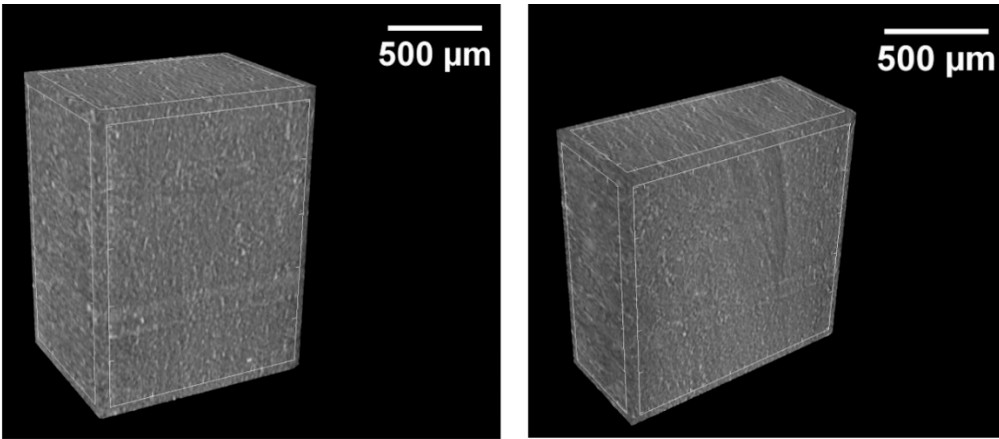

**Figure 5.** µCT scans cut from LFA samples with small (left) and large (right) particles with a filler content of 15 vol.%.

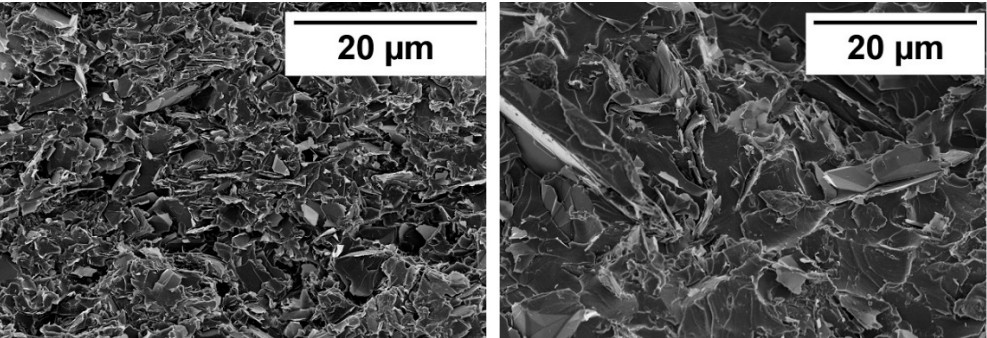

**Figure 6.** SEM of small (3.6 µm) and large (18.4 µm) graphitic particles at a magnification of 2 K. Samples from LFA have been broken manually.

*3.2. Laminate Characterization*

As shown by Rolfes, the fiber volume content has an influence on the thermal conductivity [2]. To verify the comparability of the samples, the fiber volume content has been measured and calculated by thermogravimetry. The method has been described and verified by Monkiewitsch [27]. As shown in Table 2, the fiber volume content varies only slightly between the samples, so we can expect them to be comparable.

**Table 2.** Fiber volume content of the produced laminates.

| Prepreg Laminate | Fiber Volume Content |
|---|---|
| unfilled | 48 vol% |
| Graphite ø 18.4 µm–10 vol% | 55 vol% |
| Graphite ø 18.4 µm–15 vol% | 53 vol% |
| Graphite ø 18.4 µm–20 vol% | 57 vol% |
| Graphite ø 3.6 µm–15 vol% | 56 vol% |

Figure 7 shows the thermal conductivities of the prepreg laminates measured by LFA. The unfilled resin has a thermal conductivity of 0.23 W/mK, which is increased to 0.36 W/mK by the incorporation of carbon fibers. The values are lower than those of other researchers. Rolfes and Hammerschmidt used fibers with a transverse thermal conductivity of 2 W/mK with round carbon fibers; they measured a transverse thermal conductivity of 0.708 W/mK with 64.3 vol% of fiber with the steady-state guarded hot plate (GHP) setup. For the cross-section type used in this study, Shim et al. showed a significantly lower thermal conductivity [4]. The values are also not comparable, as different measurement methods

have been used, and given that Rolfes and Hammerschmidt already showed that there is a strong variation in the measured thermal conductivity with the GHP and the THS methods [2]. When the conductive graphite filled resin with a thermal conductivity of 0.7 W/mK is used for the prepreg production, the conductivity in the laminate increases to 0.91 W/mK for the large particles.

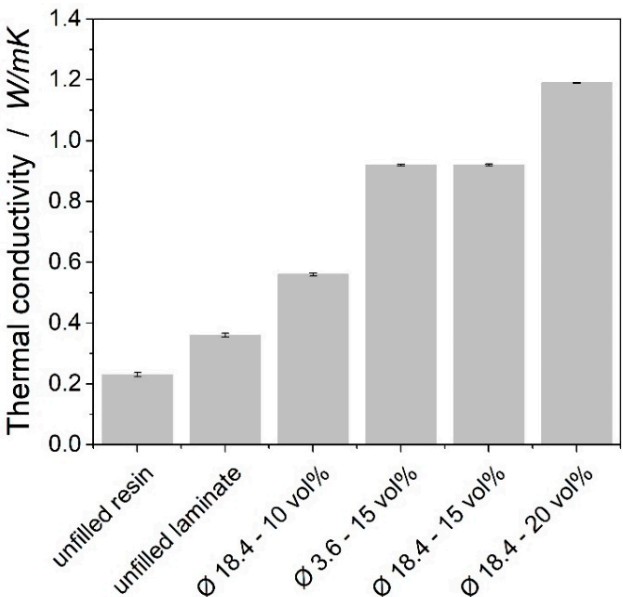

**Figure 7.** Thermal conductivity of prepreg laminates measured by LFA.

Although the results from the graphite filled resins without carbon fibers indicate that larger particles lead to higher conductivities in the composite, the effect could not be repeated with carbon fiber reinforced composites. Figure 7 shows a strong effect of the filler volume content of the laminates on the thermal conductivity, as the conductivity of the laminate significantly increases when the filler content is varied from 10 to 15 and to 20 vol%.

Figure 8 shows records from μCT of the prepreg laminates. The images show that the samples are very homogenous. From the pictures that have been evaluated, no voids could be found in the prepreg laminates. For a deeper understanding of the morphology of the prepreg laminates, optical microscopy has been used to produce and analyze polished micrograph sections. In the unfilled laminate, the fibers are well distributed. They are not absolute homogenously distributed, as resin-rich areas can be detected in the cross-section. The laminates with filled matrices have a layered structure with layers of resin and layers of fiber. As can be seen in Figure 9, the distance between the carbon fiber layer varies between 80 and 200 μm. From the evaluated images, no direct contact between the carbon fiber layers could be found.

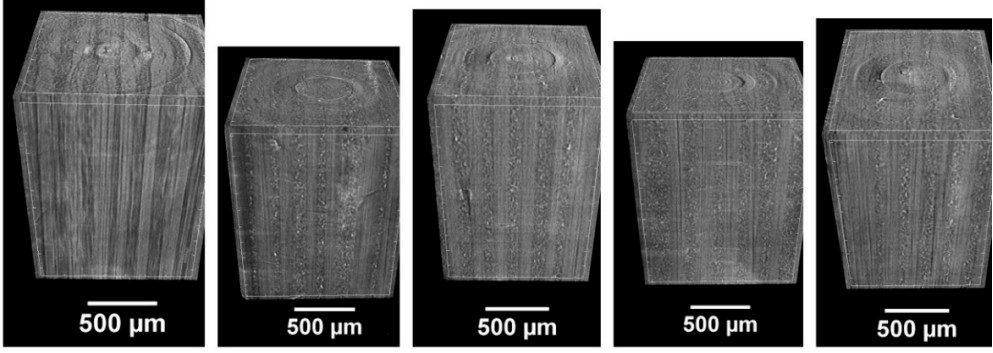

**Figure 8.** μCT scans of prepreg laminates, from left to right: unfilled, 10, 15, 20 vol% of large particles, 15 vol% of small particles.

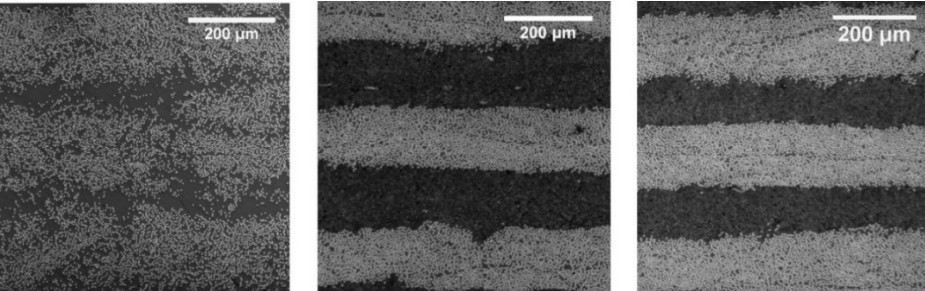

**Figure 9.** Optical microscopy records of prepreg laminates, from left to right: unfilled, 15 vol% of small particles, 15 vol% of large particles.

The smaller particles of graphite had been chosen to test the assumption that particles with a smaller diameter than the fibers could be adsorbed between the fiber layer. As shown in Figure 10, only very few particles of the graphite with 3.6 μm will be pushed between the carbon fiber layers during the prepreg production and the curing process. Most particles stay in the matrix between the layers. Almost no larger particles of 18.4 μm could be found between the carbon fibers in the laminate. It seems that the layer of carbon-fiber is very dense and hence particles are not able to be infiltrated. Also, the very high viscosity of the graphite filled epoxy resin might reduce the movability of the graphite particles.

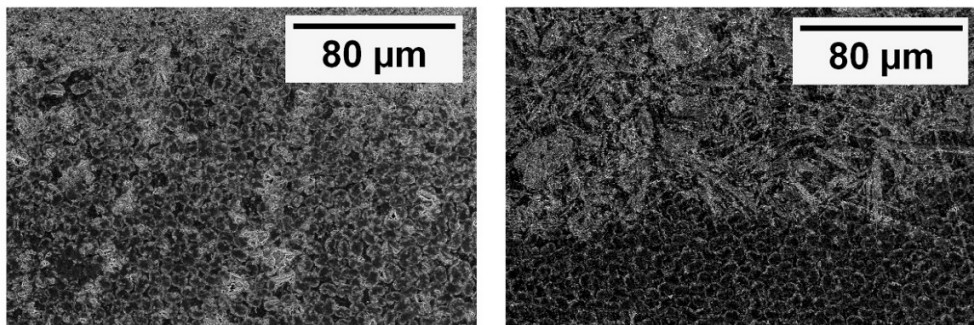

**Figure 10.** SEM records of the prepreg laminates, left: with small particles of 3.6 μm; and right: with particles of 18.4 μm.

## 4. Micromechanical Calculations

The Lewis and Nielsen's formula is used in various publications to calculate the thermal conductivity of composites [2,22,28,29]. It is formulated as follows:

$$\lambda_k = \lambda_M \frac{1 + A\,B\,\phi}{1 - B\,\phi\,C} \tag{1}$$

where $\phi$ is defined as the filler content and $\lambda_M$ as the conductivity of the matrix. The factors $A$, $B$ and $C$ reflect the filler geometry, orientation and thermal conductivity. According to Guth [30], A can be calculated with the aspect ratio $p$ of the filler:

$$A = \frac{p}{[2\ln(2p)] - 3} + 1 \tag{2}$$

$B$ is not an independent variable, since it also reflects the conductivity of the filler and matrix, and $C$ reflects the maximal packing density $\phi_{max}$:

$$B = \frac{\frac{(\lambda_F)}{(\lambda_M)} - 1}{\frac{(\lambda_F)}{(\lambda_M)} + A} \tag{3}$$

$$C = 1 + \frac{(1 - \phi_{max})}{\phi_{max}^2} \phi \tag{4}$$

The maximal packing density of the fibers in the composite is 78%, with the thermal conductivity of the matrix given above and the transverse thermal conductivity of the fiber of 1.1 W/mK. The transverse thermal conductivity of the fiber can only be estimated as no method to measure a single fiber could be identified in the literature. Rolfes and Hammerschmidt [2] calculated a transverse conductivity of 2 W/mK from their experimental data for round-type PAN fiber. For the cross-section type used in this study, Shim et al. showed a significantly lower thermal conductivity, so a thermal conductivity of 1.1 W/mK seems reasonable for this study [4]. A is calculated to 0.83 with an aspect ratio of 0.5, as suggested in the literature [2].

Figure 11 shows the experimental data and calculations with the above equation of transverse thermal conductivities of the laminates. For the unfilled laminate and the laminate with 10 vol% of graphitic filler, the Lewis/Nielsen calculation overestimates the transverse thermal conductivities. For the laminates of 15 vol% of graphite and 20 vol% of graphite, the calculations with the Lewis/Nielsen formula seem very accurate.

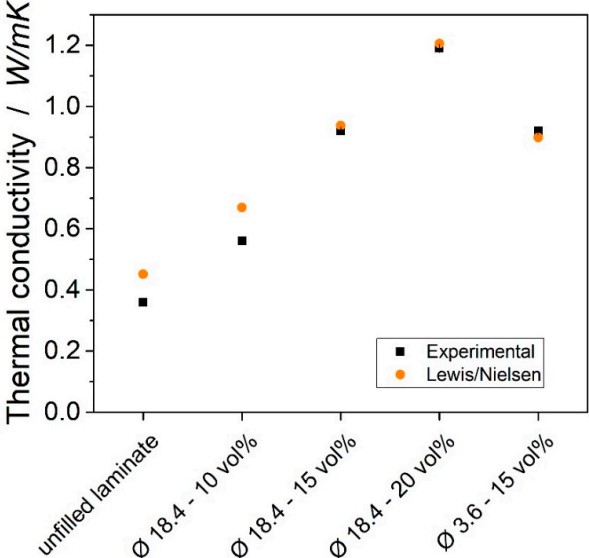

**Figure 11.** Thermal conductivity of the laminates from the experimental data and calculated according to Lewis/Nielsen equations.

## 5. Conclusions

The aim of this work was to explore the influence of a graphitic filler on the thermal conductivity of an epoxy-based carbon-fiber reinforced polymer. The following conclusions can be drawn from the analysis of the results:

- In the epoxy resin, a non-linear dependence of the thermal conductivity from the filler volume content could be found. Larger particles showed a higher conductivity, most probably due to lower phonon scattering.
- In the carbon fiber reinforced composites, the thermal conductivity was enhanced by a factor of 4 with 20 vol% of graphitic particles. The fiber reinforced laminate showed a different behavior compared to the epoxy graphite composites: Small and large particles showed very similar transverse thermal conductivities.

Further developments are needed to understand the influence of the fiber on the thermal conductivity. As stated above, no measurement method has been developed so far to measure the conductivity of carbon fibers, which makes simulations almost impossible. For a further increase in

the thermal conductivity, conductive fibers might be used. Furthermore, a deeper understanding of the mechanical properties of CFRP is crucial for the application of the materials in the industry.

**Author Contributions:** Conceptualization, S.B.; Methodology, S.B.; Validation, S.B., M.R. and V.A.; Formal Analysis, S.B. and M.R.; Investigation, M.R.; Resources, V.A.; Data Curation, S.B. and M.R.; Writing-Original Draft Preparation, S.B.; Writing-Review & Editing, M.D.; Visualization, S.B.; Supervision, S.B.; Project Administration, S.B.

**Funding:** Authors kindly thank to the German Ministry of Economy and Energy (BMWi) for the funding of the Lufo Project TELOS (FKZ 20Y1516D).

**Conflicts of Interest:** There are no conflicts of interest associated with this publication.

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
