# Peer review of "Transverse Thermal Conductivity of Epoxy Carbon Fiber Prepreg Laminates with a Graphite Filled Matrix"

_jcs, doi:10.3390/jcs3020044_

Round 1

Reviewer 1 Report

The paper is well focused and nicely presented. It can be accepted in the present form

Author Response

Thank you very much for the feedback.

Reviewer 2 Report

The paper in hand deals with enhancing the through-plane thermal conductivity of epoxy-carbon fibre cross-plied laminates by modifying the epoxy matrix by various amounts of graphite filler.

The experiments are reasonably well described and the background is well established. 

The authors find that adding graphite to the epoxy is an effective way of enhancing the through-plane thermal conductivity, outperforming other approaches—notably fibre coating— described in the literature.

From a conceptual stand-point the findings are as expected: It is well known that in composites in which the matrix is less conducting than the reinforcements (in this case the carbon fibres) improving the matrix conductivity is more effective than improving the fiber conductivity.

The observation by the authors that smaller graphite platelets would be (slightly) less efficient in improving the thermal conductivity of the epoxy compared to their larger counterparts stands also to reason, due to the lower effective thermal conductivity (taking into account the thermal resistance at the epoxy/graphite interface). Even the absence of a difference in the cross-plies with a matrix using large and small graphite particles stands to reason: at 15 vol% of graphite filler the fibre transverse thermal conductivity is about the same as that of the filled matrix and even model predictions give roughly the same thermal conductivity.

The non-linearity of the evolution of the thermal conductivity with increasing graphite volume fraction is probably linked to percolation of the (high-aspect-ratio) graphite platelets. At any rate in the low filler volume fraction range the experimental data seem to follow the linear trend as expected from non percolating systems with random filler orientation.

By and large the paper is a decent contribution, albeit not a thrilling surprise that will impact much on the science community. The engineering community however might profite from the paper.

There are a few editing changes as well as minor clarifications required before the paper can be accepted for publication:

a) clarifications:

In line 82 the authors mention "15 cm3 of graphite". Was that back-calculated from mass and density or just a volume of 15 cm3 tapped graphite powder?

Eqn. (2): what is p?

line 226: contrarily to line 95 where a "aimed for" volume fraction of 55 vol. % of fibres was indicated, the fiber volume fraction in line 226 is given as 78%. Did the authors mean the maximum fibre volume fraction?

b) on form:

typos (list might not be exhaustive):

line 35: differ significantly rather than significantly differ

line 55: ...Burger et al. concluded...

line 60: ...research is....

line 73: The density is 2.255g/cm3

line 74: The data measured...

line 107: ...were sputtered by a Cressington...

line 108: ...studied in a Jeol....

line 112: ...by the Cape-Lehman algorithm.

line 131: ...as can be observed...

line 155: ...of the µCT is 2.5 µm...

line 168: The method has been described...

References [3], [4], and [8] make reference to the same Journal but use different versions of the name. It is possible that this is taken care of by the Editing process. Otherwise it would be good if the authors settled for either the full name or the abbreviation. A similar remark is with regard to papers with more than 6 authors: the present authors have chosen to name the first 6 authors and complete with "et al." which is unusual in the context of a list of references in a Journal.

Author Response

Thank you very much for the feedback.

I reworked the paper accordingly and added the requested information.

a) clarifications:

In line 82 the authors mention "15 cm3 of graphite". Was that back-calculated from mass and density or just a volume of 15 cm3 tapped graphite powder?

Ø  I added the relevant information in the text.

Eqn. (2): what is p?

Ø  I added the relevant information in the text.

line 226: contrarily to line 95 where a "aimed for" volume fraction of 55 vol. % of fibres was indicated, the fiber volume fraction in line 226 is given as 78%. Did the authors mean the maximum fibre volume fraction?

Ø  Yes, thank you, I added this information.

b) on form:

typos (list might not be exhaustive):

line 35: differ significantly rather than significantly differ

line 55: ...Burger et al. concluded...

line 60: ...research is....

line 73: The density is 2.255g/cm3

line 74: The data measured...

line 107: ...were sputtered by a Cressington...

line 108: ...studied in a Jeol....

line 112: ...by the Cape-Lehman algorithm.

line 131: ...as can be observed...

line 155: ...of the µCT is 2.5 µm...

line 168: The method has been described...

References [3], [4], and [8] make reference to the same Journal but use different versions of the name. It is possible that this is taken care of by the Editing process. Otherwise it would be good if the authors settled for either the full name or the abbreviation. A similar remark is with regard to papers with more than 6 authors: the present authors have chosen to name the first 6 authors and complete with "et al." which is unusual in the context of a list of references in a Journal.

Reviewer 3 Report

The paper “Transverse thermal conductivity of epoxy carbon fiber prepreg laminates with graphite filled matrix” provides results about the enhancement of the transverse thermal conductivity of epoxy carbon fiber prepreg laminates through the addition of graphite fillers inside the epoxy matrix. The paper shows some interesting insights. However, I believe a few issues have to be addressed before the paper can be considered for publication:

1)   The abstract starts: “The conductivity of carbon fiber reinforced….”. The authors should specify they are talking about THERMAL conductivity.

2)   The first sentence of the introduction does not read well. Authors should rephrase and avoid starting with the word “Especially”.

3)   Table 1. Authors compared results with those of ref [5]. I wonder if there are other literature works to consider in this comparison, besides ref [5]. Also, please, use “Enhancement method” instead of “Enhancement through…”

4)   Line 121 (beginning of 3.1 section) and Figure3. Authors need to specify that they are talking about the thermal conductivity of the resin only, otherwise readers get confused

Author Response

Thank you very much for the feedback.

I reworked the paper accordingly and added the requested information.

1)   The abstract starts: “The conductivity of carbon fiber reinforced….”. The authors should specify they are talking about THERMAL conductivity.

2)   The first sentence of the introduction does not read well. Authors should rephrase and avoid starting with the word “Especially”.

3)   Table 1. Authors compared results with those of ref [5]. I wonder if there are other literature works to consider in this comparison, besides ref [5]. Also, please, use “Enhancement method” instead of “Enhancement through…”

Ø  There is more work to consider. One might add values from the literature about metal-coated fibers and mesophase pitch fiber. For this specific overview, I was looking for publications dealing with the enhancement trough matrix modification. However I added one more publication about BN-filled mesophase pitch fiber composites.

4)   Line 121 (beginning of 3.1 section) and Figure3. Authors need to specify that they are talking about the thermal conductivity of the resin only, otherwise readers get confused

Round 2

Reviewer 2 Report

The authors have satisfactorily handled the requests of the reviewer.